# SEMI-AUTOREGRESSIVE ENERGY FLOWS: TOWARDS DETERMINANT-FREE NORMALIZING FLOWS

## ABSTRACT

Normalizing flows are a popular approach for constructing probabilistic and generative models. However, maximum likelihood training of flows is challenging due to the need to calculate computationally expensive determinants of Jacobians. This paper takes steps towards addressing this challenge by introducing objectives and model architectures for determinant-free training of flows. Central to our framework is the energy objective, a multidimensional extension of proper scoring rules that admits efficient estimators based on random projections. The energy objective does not require calculating determinants and therefore supports general flow architectures that are not well-suited to maximum likelihood training. In particular, we introduce semi-autoregressive flows, an architecture that can be trained with the energy loss, and that interpolates between fully autoregressive and non-autoregressive models, capturing the benefits of both. We empirically demonstrate that energy flows achieve competitive generative modeling performance while maintaining fast generation and posterior inference.

## 1 INTRODUCTION

Normalizing flows are one of the major families of probabilistic and generative models (Rezende and Mohamed, 2015; Kingma et al., 2016; Papamakarios et al., 2019). They feature tractable inference and maximum likelihood learning, and have applications in areas such as image generation (Kingma and Dhariwal, 2018; Dinh et al., 2014), anomaly detection (Nalisnick et al., 2019), and density estimation (Papamakarios et al., 2017; Dinh et al., 2017). However, flows require calculating computationally expensive determinants of Jacobians in order to evaluate their densities; this either limits the range of architectures compatible with flows, or makes flow models with highly expressive neural architectures slow to train.

This paper seeks to question the use of maximum likelihood for training flows and instead explores an approach for determinant-free training inspired by two-sample testing and the theory of proper scoring rules. See Appendix G for more detailed motivation. Si et al. (2022) recently showed that normalizing flows can be trained using objectives derived from proper scoring rules (Gneiting and Raftery, 2007a) that involve only samples from the model and the data distribution (hence do not require computing densities and departs from log-likelihood based training of autoregressive models as in (Papamakarios et al., 2017)). Although quantile flows (Si et al., 2022) are determinant-free, they are also necessarily autoregressive due to the CDF only existing in one dimension, and therefore inherit various limitations, such as slow sampling speed.

Here, we extend the sample-based proper scoring rule framework of Si et al. (2022) to models that are not fully autoregressive. Central to our approach is the *energy objective*, a multidimensional extension of proper scoring rules that only requires model samples, and not densities. We complement this objective with efficient estimators based on random projections and compare against alternative sample-based objectives that serve as strong baselines. We examine the theoretical properties of our approach, draw connections to divergence minimization, and highlight benefits over maximum likelihood training.

Our framework enables training model architectures that are more general than the ones compatible with maximum likelihood learning (e.g., densely connected networks). In particular, we propose semi-autoregressive flows, an architecture trained with the energy loss that integrates the speed of feed-forward architectures with the sample quality of autoregressive models. Across a number

Table 1: Energy Flows and Semi-Autoregressive Energy Flows (SAEFs) are invertible generative models that feature expressive architectures, exact likelihood and posterior evaluation, and their training does not require computing log-determinants, in contrast to VAEs (Kingma and Welling, 2014), MAFs (Papamakarios et al., 2017), NAFs (Huang et al., 2018), AQFs (Si et al., 2022), GMMNets (Li et al., 2015), and CramerGANs (Bellemare et al., 2017).

| Method | Likelihood | Posterior | Sampling | Representation | Objective |
|---|---|---|---|---|---|
| VAE | Approx. | Approx. | Feedforward | Gaussian | ELBO |
| MAF | Exact | Exact | Autoregressive | Gaussian | Likelihood |
| NAF | Exact | Exact | N/A | Neural | Likelihood |
| AQF | Exact | Exact | Autoregressive | Neural | Quantile Loss |
| GMMNet | N/A | N/A | Feedforward | Neural | MMD |
| CramerGAN | N/A | N/A | Feedforward | Neural | Discr.+Energy |
| Energy Flow (Ours) | Exact | Exact | Feedforward | Neural | Energy Loss |
| SAEF (Ours) | Exact | Exact | Autoregressive | Neural | Energy Loss |

of generative modeling tasks, our method produces high quality samples, while supporting exact posterior inference, which we demonstrate via an exploration of the models' latent space. Table 1 compares our approach to existing methods.

**Contributions.** In summary, this work (1) questions the use of maximum likelihood for training flows and proposes an alternative approach based on proper scoring rules and two-sample tests that extends quantile flows (Si et al., 2022) to multiple dimensions. We (2) introduce specific two-sample objectives, such as the energy loss, and derive efficient slice-based estimators. We also (3) provide a theoretical analysis for the proposed objectives as they are consistent estimators and feature unbiased gradients. Finally, we (4) introduce a semi-autoregressive architecture that features high sample quality and speed on generation and posterior inference tasks.

## 2 BACKGROUND

**Normalizing Flow Models** Generative modeling involves specifying a probabilistic model $p(\mathbf{y}) \in \Delta(\mathbb{R}^d)$ over a high-dimensional $\mathbf{y} \in \mathbb{R}^d$ (Kingma and Welling, 2014; Goodfellow et al., 2014). A normalizing flow is a generative model $p(\mathbf{y})$ defined via an invertible mapping $f : \mathbb{R}^d \to \mathbb{R}^d$ between a noise variable $\mathbf{z} \in \mathbb{R}^d$ sampled from a prior $\mathbf{z} \sim p(\mathbf{z})$ and the target variable $\mathbf{y}$ (Rezende and Mohamed, 2015; Papamakarios et al., 2019). We may obtain an analytical expression for the likelihood $p(\mathbf{y})$ via the change of variables formula $p(\mathbf{y}) = \left| \frac{\partial f(\mathbf{z})^{-1}}{\partial \mathbf{z}} \right| p(\mathbf{z})$, where $\left| \frac{\partial f(\mathbf{z})^{-1}}{\partial \mathbf{z}} \right|$ denotes the determinant of the inverse Jacobian of $f$. Computing this quantity is often expensive, hence we typically choose $f$ to be in a class of models for which the determinant is tractable (Rezende and Mohamed, 2015), such as in autoregressive models (Papamakarios et al., 2017).

**Proper Scoring Rules** Consider a score or a loss $\ell : \Delta(\mathbb{R}^d) \times \mathbb{R}^d \to \mathbb{R}_+$ over a probabilistic forecast $F \in \Delta(\mathbb{R}^d)$ and a sample $\mathbf{y} \in \mathbb{R}^d$. The loss $\ell$ is proper if the true distribution $G \in \arg\min_F \mathbb{E}_{\mathbf{y} \sim G} \ell(F, \mathbf{y})$ (Gneiting and Raftery, 2007a). A popular proper loss is the continuous ranked probability score (CRPS), defined for two cumulative distribution functions (CDFs) $F$ and $G$ as $\mathrm{CRPS}(F, G) = \int (F(y) - G(y))^2 \, dy$. When we only have samples from $G$, we can generalize this score to obtain the following loss for a single sample $y'$: $\mathrm{CRPS}_s(F, y') = \int_y (F(y) - \mathbb{I}(y - y'))^2 \, dy$. where $\mathbb{I}$ denotes the Heaviside step function. The above CRPS can also be written as an expectation relative to the distribution $F$:

$$\mathrm{CRPS}(F, y') = -\frac{1}{2} \mathbb{E}_F |Y - Y'| + \mathbb{E}_F |Y - y'|, \tag{1}$$

where $Y, Y'$ are independent copies of a random variable distributed according to $F$. Recently, Si et al. (2022) proposed autoregressive quantile flows, which are trained using the CRPS and are determinant-free. We seek to extend the approach of Si et al. (2022) beyond autoregressive flows.

**Two-Sample Tests and Integral Probability Metrics** Two-sample tests compare distributions $F, G$ based on their respective sets of samples $\mathcal{D}_F = \{\mathbf{y}^{(i)}\}_{i=1}^m$ and $\mathcal{D}_G = \{\mathbf{x}^{(i)}\}_{i=1}^n$. Specifically,

a two-sample test defines a statistic $T : \mathbb{R}^d \to \mathbb{R}$; we determine whether $\mathcal{D}_F, \mathcal{D}_G$ originate from identical or different distributions $F, G$ based on differences in $T$ across $\mathcal{D}_F, \mathcal{D}_G$. Two-sample tests motivate objectives for generative models such as generative moment matching networks (GMMNs; (Dziugaite et al., 2015; Li et al., 2015)) and generative adversarial networks (GANs; (Goodfellow et al., 2014)). Two-sample tests are also an attractive training objective for flows: because they are density-free, they do not require computing determinants (Grover et al., 2018).

More modern approaches include integral probability metrics (IPMs) (Müller, 1997), which take the form $\max_{T \in \mathcal{T}} \mathbb{E}_{y \sim F}[T(y)] - \mathbb{E}_{y \sim G}[T(y)]$, where $\mathcal{T}$ is a family of functions. A special case of IPMs is maximum mean discrepancy (MMD) (Gretton et al., 2008), in which $\mathcal{T} = \{T : ||T||_{\mathcal{H}} \leq 1\}$ is the set of functions with bounded norm in a reproducing kernel Hilbert space (RKHS) with norm $|| \cdot ||_{\mathcal{H}}$; the CRPS objective can be shown to be a form of MMD (Gretton et al., 2008).

## 3 EXPLORING DETERMINANT-FREE TRAINING OF NORMALIZING FLOWS

We propose training normalizing flows using objectives inspired by two-sample tests, which do not require computing densities. This idea poses two sets of challenges: (1) most classical two-sample tests (e.g., Kolmogorov-Smirnov) are defined in one dimension and do not have simple multivariate extensions; (2) modern two-sample tests (e.g., IPMs) extend to high dimensions, but typically require solving a costly optimization problem. Here, we derive two-sample tests that form good learning objectives, and we use the theory of proper scoring rules to justify their validity.

### 3.1 SAMPLE-BASED TRAINING OF NORMALIZING FLOWS AND THE ENERGY OBJECTIVE

We seek to extend autoregressive quantile flows (Si et al., 2022) to a general structure without the limitations of autoregressivity (e.g., slow sampling). Specifically, we leverage a generalization of the sample-based form of the CRPS objective (1) to a multi-dimensional version called the *energy score* by Szekely (2003); Gneiting and Raftery (2007a):

$$\text{CRPS}_e(F, \mathbf{y}') = -\frac{1}{2}\mathbb{E}_F||\mathbf{Y} - \mathbf{Y}'||_{\mathbf{2}}^{\beta} + \mathbb{E}_F||\mathbf{Y} - \mathbf{y}'||_{\mathbf{2}}^{\beta}, \tag{2}$$

where $\beta \in (0, 2)$, $|| \cdot ||_{\mathbf{2}}$ denotes the Euclidean norm, and $\mathbf{Y}, \mathbf{Y}' \in \mathbb{R}^d$ are independent copies of a vector-valued random variable distributed according to $F$. The rightmost term $\mathbb{E}_F||\mathbf{Y} - \mathbf{y}'||_{\mathbf{2}}^{\beta}$ promotes samples $\mathbf{Y}$ from $F$ that are close to the data point $\mathbf{y}'$; the leftmost term $\frac{1}{2}\mathbb{E}_F||\mathbf{Y} - \mathbf{Y}'||_{\mathbf{2}}^{\beta}$ encourages the model to produce diverse samples and not concentrate all probability mass on one $y$.

**The Kernelized Energy Objective**   As a generalized extension of the CRPS, the Kernelized Energy Objective extends the Euclidean norm to a kernel function:

$$\text{CRPS}_K(F, \mathbf{y}') = -\frac{1}{2}\mathbb{E}_F K(\mathbf{Y}, \mathbf{Y}') + \mathbb{E}_F K(\mathbf{Y}, \mathbf{y}'), \tag{3}$$

The kernelized energy loss can be shown to be a proper loss (Gneiting and Raftery, 2007a), and thus represents a valid training objective for a generative model. The flow objective consists of $\mathbb{E}_{\mathbf{y}' \sim \mathcal{D}}[\text{CRPS}_K(F, \mathbf{y}')]$ and also reveals the connection to two-sample tests between $F$ and $\mathcal{D}$.

**Two-Sample Baselines**   In Appendix D.1, We define two classical statistical tests as baselines and to illustrate examples of alternative methods that can be derived from our two-sample-based approach. In brief, **Hotelling's two-sample test** uses the statistic $H_2(\mathcal{D}_F, \mathcal{D}_G) = (\mathbf{m}_F - \mathbf{m}_G)^\top S^{-1}(\mathbf{m}_F - \mathbf{m}_G)$, where $\mathbf{m}_F, \mathbf{m}_G$ are sample means, $S_F, S_G$ are sample variances and $S = (S_F + S_G)/2$. The **Fréchet distance** uses the objective $R(\mathcal{D}_F, \mathcal{D}_G) = ||\mathbf{m}_F - \mathbf{m}_G||_2^2 + \text{tr}(S_F + S_G - 2(S_F S_G)^{1/2})$, where we are using the same notation.

### 3.2 THEORETICAL PROPERTIES

**Divergence Minimization**   When the variable $y \in \mathbb{R}$ is one-dimensional, the objective $\text{CRPS}(F, G)$ is precisely equivalent to the Cramér divergence $\ell_2^2(F, G) = \int_{-\infty}^{\infty}(F(y) - G(y))^2 dy$ between distributions $F, G$. Szekely (Szekely, 2003) showed that the one-dimensional version of the energy

loss (2) is precisely equivalent to $\ell_2^2$. Its kernelized version can be shown to also represent a valid divergence between distributions (Gneiting and Raftery, 2007a). The connection to divergence minimization lends additional support to using (2), (3) as principled objectives—if $G$ is the data distribution, minimizing (2), (3) over a space of models will produce a model $F$ that is close to $G$.

**Unbiased Gradient Estimation**    Our objectives have the property that given a sequence $\mathbf{Y}_K$ of $K$ samples $\mathbf{y}_1, \mathbf{y}_2, ..., \mathbf{y}_K$ from $G$, the gradient of the empirical distribution over these samples yields an unbiased estimate of the gradient of the expected loss: $\nabla_\theta \mathbb{E}_{\mathbf{Y}_K} \ell(F_\theta, \hat{G}_K) = \nabla_\theta \ell(F_\theta, G)$, where $\ell$ is one of our objective functions, $\hat{G}_K$ is the empirical distribution over $\mathbf{Y}_K$ and $F_\theta$ is a model with parameters $\theta$ that we are optimizing. The above fact follows directly from the fact that both the energy and the CRPS objectives are proper scoring rules (Bellemare et al., 2017).

**Why Energy Objectives?**    Consider the set of objectives $\ell_p^p(F, G) = \int_{-\infty}^{\infty} (F(y) - G(y))^p dy$ for $p \geq 1$ over $y \in \mathbb{R}$; these are also known as Wasserstein $p$-metrics (Kantorovich, 1939). The energy objective corresponds to $\ell_2^2$, and it is *the only* $\ell_p^p$ objective to support unbiased gradients (Bellemare et al., 2017). In high dimensions, IPMs are general-purpose two-sample tests; popular IPMs include the Kantorovich metric (Kantorovich and Rubinstein, 1958), Fortet-Mourier metric (Fortet and Mourier, 1953), the Lipschitz (or Dudley) metric (Dudley, 1966), and the total variation distance. In general, IPMs are defined in terms of a potentially costly optimization problem; out of the aforementioned IPMs, only the energy objective has a known analytical (optimization-free) solution, and it also features a faster statistical convergence rate (Sriperumbudur et al., 2009).

Overall, we summarize the above facts as part of the following formal result.

**Theorem 1.** *The energy objectives (2) and (3) are consistent estimators for the data distribution and feature unbiased gradients.*

This follows from properties of proper scoring rules and MMD; see the Appendix for a full proof.

### 3.3    Scaling Sample-Based Flow Objectives Using Projections

The framework of IPMs provides a wide range of high-dimensional sample-based objectives (Müller, 1997). However, most of these objectives involve costly optimization problems, with the energy loss being a rare exception. At the same time, there exist many popular one-dimensional two-sample tests that have appealing statistical and computational properties and can yield training objectives.

We propose further improving our objective via *random projections*, specifically *slicing*, which projects data into one dimension (Kolouri et al., 2019; Song et al., 2019; Nguyen et al., 2020). Formally, we define a sampling probability $p(\mathbf{v})$ over one-dimensional vectors $\mathbf{v} \in \mathbb{R}^d$. We define a sliced version of a one-dimensional loss function $L(x, y) : \mathbb{R} \times \mathbb{R} \to \mathbb{R}$ as

$$L_p(\mathbf{x}, \mathbf{y}) = \mathbb{E}_{\mathbf{v} \sim p(\mathbf{v})} \left[ L(\mathbf{v}^\top \mathbf{x}, \mathbf{v}^\top \mathbf{y}) \right]. \tag{4}$$

We approximate the expectation with a number of Monte-Carlo samples.

**Sliced Energy Objectives**    The sliced energy objective applies Equation 1 to the projected data. In practice, we find that the number of slices needed for good performance is lower than the dimensionality of the data, resulting in a favorable computational profile. Furthermore, we can formally prove that the resulting objective has appealing statistical properties.

**Theorem 2.** *The sliced versions of the energy objectives (2) and (3) are consistent estimators for the data distribution and feature unbiased gradients.*

Intuitively, the first part of the theorem is true because the CRPS objective is related to the MMD. At the same time, for each $\mathbf{v}$ the objective remains a proper score; a weighted combination of proper scores is also a proper score, hence the second part holds. See the Appendix for a full proof. Recall also that in one dimension, the energy loss reduces to the CRPS, which is equivalent to the Wasserstein-2 distance. Wasserstein distances have more favorable convergence properties (Arjovsky et al., 2017) than maximum likelihood training, which lends further support to our choice of objective.

**Sliced Two-Sample Baselines** Slicing also allows us to use univariate two-sample tests as objectives. We describe several objectives in Appendix D.2. In brief, these include **Kolmogorov-Smirnov**, a popular statistical test defined for two CDFs $F$ and $G$ as $\mathrm{KS}(F,G) = \sup_y |F(y) - G(y)|$; **Hotelling's** $t^2$ test $H_u(\mathcal{D}_F, \mathcal{D}_G) = \frac{(m_F - m_G)^2}{s^2}$, a sliced version of Hotelling's objective; the sliced version of the **Fréchet** objective $R_u(\mathcal{D}_F, \mathcal{D}_G) = (m_F - m_G)^2 + (s_F^2 - s_G^2)^2$.

## 4 ARCHITECTURES FOR ENERGY FLOWS

Next, we introduce *energy flows*, a class of models trained with our proposed determinant-free objectives. An energy flow is defined by an invertible mapping between $z$ and $y$ and is trained using the energy loss. As a result, energy flows improve over classical flow models by, among other things, supporting flexible architectures and by simultaneously providing fast training and sampling.

Previous work on normalizing flows involved constrained architectures with tractable determinants, such as autoregressive models. In contrast, our model supports flexible feedforward architectures, which we outline in Appendix E. In brief, our loss supports **dense invertible flows** (DIFs), sequences of fully connected layers constrained to be invertible, **invertible residual networks** (Behrmann et al., 2019), as well as **rectangular flows** (REFs), in which the dimensionality of $y$ and $z$ is not equal (Nielsen et al., 2020; Cunningham and Fiterau, 2021; Caterini et al., 2021).

### 4.1 SEMI-AUTOREGRESSIVE FLOWS

We also introduce semi-autoregressive flows (SAEFs), an architecture trained with the energy loss that combines the speed of feed-forward architectures with the sample quality of autoregressive models. The SAEF model divides $d$-dimensional the data into $B$ blocks and generates samples blockwise; as a result, sampling time is reduced by a factor of $O(d/B)$ relative to an autoregressive model.

Formally, SAEFs define an invertible mapping between a latent variable $\mathbf{z}$ and an observed variable $\mathbf{y}$ and require choosing a partition of $\mathbf{y}, \mathbf{z}$ into $B$ ordered blocks $(\mathbf{y}_b)_{b=1}^B$ and $(\mathbf{z}_b)_{b=1}^B$ (e.g., 4x4 blocks of pixels in an image). They induce a probabilistic model $p(\mathbf{y})$ over $\mathbf{y}$ that factorizes as $p(\mathbf{y}) = \prod_{b=1}^B p(\mathbf{y}_b|\mathbf{y}_{<b})$, where each $p(\mathbf{y}_b|\mathbf{y}_{<b})$ is defined via an invertible mapping

$$\mathbf{y}_b = \tau(\mathbf{z}_b; \mathbf{h}_b) \qquad\qquad \mathbf{h}_b = c_b(\mathbf{y}_{<b}), \qquad\qquad (5)$$

where $\tau(\mathbf{z}_b; \mathbf{h}_b)$ is an invertible transformer mapping the $b$-th latent block $\mathbf{z}_b$ to the $b$-th observed block $\mathbf{y}_b$, and $c_b$ is the $b$-th conditioner which outputs transformer parameters $\mathbf{h}_b$. Any invertible feed-forward energy flow can be used to parameterize $\tau$—we provide specific examples below. The entire SAEF is trained via a sum of energy losses applied to each block $\mathbb{E}_{\mathbf{y}\sim\mathcal{D}}\left[\sum_{b=1}^B \ell(F_{\mathbf{y}_b}, \mathbf{y}_b)\right]$, where $\mathcal{D}$ is a training set, $\mathbf{y} \sim \mathcal{D}$ is a datapoint sampled from $\mathcal{D}$, $F_{\mathbf{y}_b}$ is the distribution over $\mathbf{y}_b$ induced by $\tau(\cdot, \mathbf{h}_b(\mathbf{y}_{<b}))$, and $\ell$ is one of our two-sample losses, such as (2) or (3).

When blocks are one-dimensional, this reduces to a standard autoregressive architecture that features high sampling quality but slow sampling speed. When blocks are full-dimensional, this reduces to a non-autoregressive energy flow with fast sampling but possibly worse quality. In our experiments, we show that SAEFs can trade-off between these two regimes and obtain the best of both worlds. Note also that SAEFs are hard to train using maximum likelihood, as they require specifying invertible non-autoregressive mappings $\tau$ between possibly high-dimensional blocks $\mathbf{y}_b, \mathbf{z}_b$—**the SAEF architecture is only trainable using the energy objective**. See Appendix F for pseudocode.

## 5 EXPERIMENTS

We evaluate our framework on a range of UCI datasets (Dua and Graff, 2017) as well as datasets of handwritten digits (Pedregosa et al., 2011; Deng, 2012).

**Classical Normalizing Flows** We benchmark our models using normalizing flows trained using maximum likelihood and that use two types of architectures: autoregressive and non-autoregressive. Our autoregressive models are based on baselines from earlier work (Papamakarios et al., 2017; Si et al., 2022) and include Neural Autoregressive Quantile Flows (NAQF), and Masked Autoregressive

Table 2: Objective Results and Complexity.

| Metrics | KS | 1D Hotelling | Hotelling | 1D-FD | FD | 1D-Energy | Energy |
|---|---|---|---|---|---|---|---|
| CRPS | 1.53 | 0.57 | 0.717 | 0.558 | 0.559 | 0.545 | 0.548 |
| Complexity | $n \log b$ | $n$ | $d^3$ | $n$ | $d^3$ | $bn$ | $bd$ |

Flows (MAF-LL). These models assume a parameterization $p(\mathbf{y}|\mathbf{z}) = \prod_{j=1}^{d} p(y_j|y_{<j}, z_j)$, where each $p(y_j|y_{<j})$ is a probability conditioned on the previous variables and $z_j$. In MAFs, the $p(y_j|y_{<j})$ are Gaussian; in NAQFs they are parameterized by a flexible quantile flow (Si et al., 2022). In our UCI experiments, this we used densely connected networks; in our image experiments, we used a variant of the PixelCNN architecture.

Our non-autoregressive models consist of variational auto-encoders (VAEs) trained using the evidence lower bound (ELBO) on the maximum likelihood and based on a fully-connected architecture (see below for details). In order to understand the benefits of our objective, we fit a VAE model with the same invertible architecture for the generator as the one used by our energy flow models; we refer to the resulting method as a dense invertible flow trained using maximum log-likelihood (DIF-LL). We additionally compare against two state-of-the-art flow models, FFJORD (Grathwohl et al., 2018) and Invertible Resnets (Behrmann et al., 2019), using their open-source codebase.

**Energy Flow Models**  We constructed flow models using dense invertible layers, referring the resulting model as a Dense Invertible Flow trained with an energy loss (DIF-E). The DIF-E model consists of three feedforward invertible layers and Leaky ReLU activation functions, and is trained using the kernelized energy loss 3 with a mixture of RBF kernels with bandwidth in $\{2, 5, 10, 20, 40, 80\}$. We also use the energy score to train non-invertible rectangular flows trained with the energy loss (REF-E). In particular, rectangular flows are parametrized by layers of size $[d/8, d/4, d/2, d]$ as compared to DIF-E which requires all layers to be of size d for invertibility.

We also compared DIF-E to autoregressive models trained with a Jacobian-free objective. Specifically, we trained our autoregressive MAF models with the quantile loss (Si et al., 2022) in addition to the NAQF models. We denote these as MAF-QL and AQF-QL respectively.

**Metrics**  We evaluate the models in terms of log-likelihood (when available and appropriate) and using variants of the CRPS metric. For VAE-type models trained using the ELBO, we report the ELBO as a lower bound on the log-likelihood. We use two CRPS-style metrics which have the following structure: the first is a sample-based version as in Equation 2. The second, marked as univariate CRPS (U-CRPS), is the sum of one-dimensional CRPS measured for each output dimension and estimates the quality of marginal distributions (Si et al., 2022). Both versions use the $\ell_1$ norm.

In our image datasets, we are also interested in estimating in a quantitative way the quality of the generated samples and their similarity to the data distribution. In order to do that, we define a metric called the D-loss. The D-Loss is measured by the accuracy of a discriminative model to determine whether an image is generated. Details can be found in the appendix.

## 5.1 UNDERSTANDING THE SAMPLE-BASED OBJECTIVES

We start with experiments that analyze the properties of the sliced energy objectives and compare them to the baseline two-sample objectives on the UCI and image datasets.

**Energy Objective vs. Two-Sample Baselines**  We claim that the energy objective is a particularly favorable training criterion for flows; we empirically establish this fact by comparing it against the other two-sample objectives, which we see as strong baselines. We train an invertible flow model on the Miniboone UCI dataset. Complexity is written where $b$ denotes the batch size, $d$ denotes the dataset dimension, and $n$ denotes the number of projections made.

Flows trained using the energy objective achieve the best performance in Table 2, outperforming the strong baselines; we focus on the energy objective in subsequent in-depth experiments.

**Improvements in Scalability From Slicing**  Next, we seek to understand the scalability improvements from slicing. We train a projection based model on MNIST using projected energy loss. The model consists of 4 dense layers of size 784 with leaky relu activation functions for the first three layers and a sigmoid activation function for the last. When we calculate the loss, we take n projections into a

Table 3: Slicing on MNIST.

| n | 400 | 200 | 100 | 50 |
|---|---|---|---|---|
| U-CRPS | 0.088 | 0.088 | 0.088 | 0.091 |
| CRPS | 0.191 | 0.191 | 0.192 | 0.195 |
| Block-n | 100 | 20 | 10 | 5 |
| U-CRPS | 0.084 | 0.085 | 0.084 | 0.084 |
| CRPS | 0.086 | 0.087 | 0.086 | 0.087 |

single dimension, which we denote in the top half of Table 3 as n for the projection parameters. We additionally conduct slicing experiments on the SAEFs, in particular the 7x7 block size variant. Slicing is conducted separately for each block in a similar manner to the fully-feedforward flow, so we're projecting a 49 dimensional block down to a single dimension. The number of projections per block is denoted in Table 3 as Block-n. In general, even for the block parameterization, the CRPS tends to stay stable when using fewer projections.

We see in Table 3 that sliced objectives perform comparably to non-sliced objectives with a fraction of the dimension, while having improved computational complexity.

**On Likelihood vs. Non-Likelihood Based Losses**  Our work questions the use of maximum likelihood for training flow models. In Table 4, we observe that a PixelMAF model trained with likelihood has low CRPS scores, and conversely it achieves poor likelihoods when trained with an energy loss (samples have high quality in both scenarios). An non-autoregressive DIF-E model achieves the highest quality samples and the worst likelihood when trained with the energy loss, suggesting that the energy loss is an attractive alternative to the likelihood, featuring high quality without the instability of GAN-type objectives (Grover et al., 2018).

## 5.2  UCI EXPERIMENTS

Non-autoregressive energy flows can capture joint dependencies without the slow generation process of autoregressive models. We implemented energy flows on five UCI datasets used previously as benchmarks by Papamakarios et al. (2017) and Si et al. (2022), which are: BSDS 300, Miniboone,

Table 4: Metrics on MNIST.

| Models | NLL | CRPS | FID |
|---|---|---|---|
| PixelMAF-LL | -3142 | .279 | 100.15 |
| PixelMAF-QL | -2617 | .215 | 85.08 |
| DIF-E | 2388 | .186 | 22.76 |

Gas, Power and Hepmass. The size of the datasets are noted in Table 5. We used an LSTM architecture for all autoregressive models and the SAEFs. The SAEFs use an appropriate block size $b$ which divides the dimension of the data; Miniboone has an extra dummy dimension appended to make it divisible by 2 in this case.

Table 5: Model performance on UCI datasets as measured by CRPS.

| Dataset | $d$ | MAF-LL | MAF-QL | AQF-QL | DIF-E | DIF-E Proj | b | SAEF |
|---|---|---|---|---|---|---|---|---|
| BSDS 300 | 63 | .044 | .036 | **.033** | .039 | .040 | 3 | .037 |
| Miniboone | 43 | .567 | .561 | .525 | .524 | .545 | 2 | **.521** |
| Gas | 8 | .645 | .565 | **.513** | .548 | .551 | 2 | .530 |
| Power | 6 | .542 | .506 | .502 | .451 | .454 | 2 | **0.443** |
| Hepmass | 21 | .617 | .614 | **.523** | .589 | .587 | 3 | .559 |

**Results.**  As shown in Table 5, energy flows perform comparably to the neural AQF-QL baseline. Both methods are trained using variants of the CRPS and obtain top performance across the five datasets. However, the DIF-E model is non-autoregressive, hence provides advantage in terms of sampling speed. Our experiments thus illustrates that non-autoregressive models can match the performance of autoregressive models trained with log-likelihood or versions of the CRPS objective. SAEFs, which use a variant of the same loss as DIF-E, further improve upon its results, giving slightly better CRPS scores across the board, while still being reasonable in terms of sampling speed.

Table 6: Digits Generation Experiments

| Method | U-CRPS | CRPS | D-Loss | Training (sec) | Sampling (sec) |
|---|---|---|---|---|---|
| PixelMAF-LL | 0.136 | 0.206 | 0.974 | 0.15 | 14.00 |
| PixelMAF-QL | 0.131 | 0.204 | 0.883 | 0.15 | 14.00 |
| PixelAQF-QL | 0.127 | 0.199 | **0.681** | 0.13 | 14.00 |
| DIF-LL (VAE) | 0.138 | 0.207 | 0.941 | 0.07 | 0.01 |
| REF-E | 0.127 | 0.201 | 0.823 | 0.12 | 0.06 |
| DIF-E | **0.126** | **0.197** | 0.807 | 0.14 | 0.07 |
| DIF-E-Proj | 0.127 | 0.199 | 0.815 | 0.06 | 0.08 |
| SAEF-1 | 0.126 | 0.198 | 0.795 | 0.12 | 10.54 |
| SAEF-2 | 0.126 | 0.199 | 0.754 | 0.07 | 2.73 |
| SAEF-4 | 0.127 | 0.198 | 0.772 | 0.05 | 0.86 |

## 5.3 Image Generation on Digits

Next, we test our methods on a standard generative modeling task: digit generation (Pedregosa et al., 2011) in Table 6. We use a PixelCNN architecture for the autoregressive models, which we denote PixelMAF-LL, PixelMAF-QL, and PixelAQF-QL. The PixelCNN maps from a $(\mathbb{N}(0,1))^d$ distribution (PixelMAF) and a $(\mathbb{U}(0,1))^d$ distribution (PixelAQF-QL) to the target distribution. SAEF models also utilize the same PixelCNN architecture, but with its generation sectioned off into different blocks, each of which is evaluated by our energy loss. We provide the ELBO loss as an upper bound on the NLL for VAE-type models, and training speed is given by seconds per epoch, while sampling speed is given by seconds per 1000 samples generated.

**Results.** The proposed DIF-E and SAEF models perform comparably on the U-CRPS and CRPS metrics to the AQF model, although the latter is more discriminable (has a better D-loss). On the other hand the samples generated by the DIF-E and SAEF outperforms those of any of the other autoregressive architectures, as well as the samples from non-autoregressive DIF-LL model, which is trained with maximum log likelihood.

**Inversion and Interpolation.** A key feature of the DIF-E model is exact posterior inference (despite not being trained with log-likelihood). To demonstrate this, we create intermediate representations between pairs of digits through which we can smoothly interpolate (Figure 1).

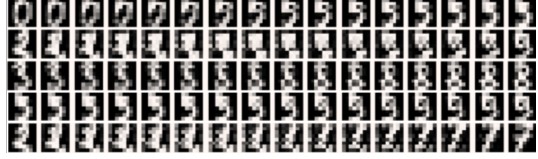

Figure 1: Interpolated digits with Energy Flow

By taking the inverse of the decoder model by inverting the activation functions and weight matrices, we are able to create an encoder similar to that of the VAE. Like the VAE, the energy flow can generate interpolated samples, with the added advantage of exact posterior inference.

## 5.4 MNIST

For this set of experimental baselines, we once again trained autoregressive models with the PixelCNN architecture, with a receptive field of 7 and 3 residual blocks. Our SAEF models use the same architecture, but with the corresponding block sizes and trained using the block energy loss. In addition, we trained a rectangular flow model using the ELBO approximation of the log-likelihood (REF-LL), which results in a model equivalent to a VAE. We computed the same metrics as we did for the experiments on the digits dataset for all methods.

**Results.** Though the fully feedforward DIF-E and DIF-E-Proj variants still showed decent performance with fast training and generation speed and show great results in CRPS, SAEF-4 in particular greatly outperforms them in FID without losing much ground in terms of CRPS, and in general by introducing a degree of autoregressivity, we see much improved results. In addition, the sampling time and complexity is still very reasonable in comparison to the fully autoregressive models, and medium size SAEFs improve upon the sampling time by an order of magnitude.

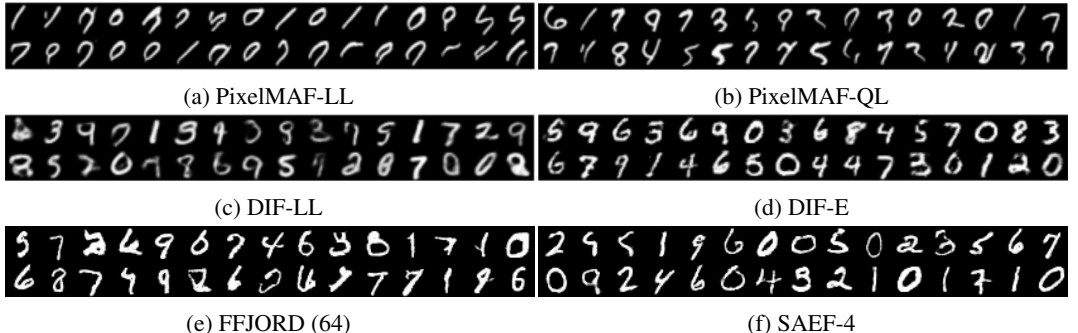

(a) PixelMAF-LL      (b) PixelMAF-QL

(c) DIF-LL      (d) DIF-E

(e) FFJORD (64)      (f) SAEF-4

Figure 2: MNIST samples from six methods

Table 7: MNIST Generation Experiments

| Method | U-CRPS | CRPS | D-Loss | FID | MMD | Training (sec) | Sampling (sec) |
|---|---|---|---|---|---|---|---|
| PixelMAF-LL | .128 | .279 | 1.00 | 100.55 | 0.296 | 35 | 195.38 |
| PixelMAF-QL | .099 | .215 | .983 | 85.08 | 0.287 | 35 | 195.38 |
| PixelAQF-QL | .119 | .228 | .986 | 61.27 | 0.232 | 29 | 195.38 |
| REF-LL (VAE) | .090 | .189 | .903 | 44.55 | 0.140 | 8 | 0.25 |
| DIF-LL (VAE) | .089 | .190 | .855 | 36.91 | 0.131 | 10 | 0.48 |
| FFJORD (16) | .101 | .208 | .650 | 24.78 | 0.103 | 540 | 48.88 |
| FFJORD (64) | .102 | .209 | .633 | 9.69 | 0.087 | 3100 | 155.69 |
| iResNet | .100 | .206 | .642 | 41.47 | 0.111 | 840 | 2.43 |
| Flow-GAN | .085 | .187 | 0.608 | 43.67 | 0.068 | 15 | 0.40 |
| GLOW | 0.090 | 0.197 | 0.983 | 63.06 | 0.600 | 1400 | 190.63 |
| REF-E | .085 | .187 | .778 | 41.04 | 0.052 | 3 | 0.21 |
| DIF-E | **.084** | **.186** | .701 | 22.76 | **0.051** | 6 | 0.40 |
| DIF-E-Proj | .085 | .186 | .819 | 22.55 | 0.056 | 3 | 0.40 |
| SAEF-2 | .085 | .188 | .675 | 9.86 | 0.167 | 32 | 31.22 |
| SAEF-4 | .085 | .187 | **.567** | **7.05** | 0.081 | 12 | 8.19 |
| SAEF-7 | .085 | .187 | .608 | 14.91 | 0.088 | 6 | 2.17 |
| SAEF-14 | .085 | .187 | .650 | 19.57 | 0.068 | 5 | 0.93 |

## 6   Previous Work, Discussion, and Conclusion

**Comparison to Other Generative Model Families**   We provide a complete comparison to other models in Table 1. Our approach contrasts against VAE style models (Kingma and Welling, 2014) by providing exact inference and likelihood evaluation. Unlike MAFs (Papamakarios et al., 2017), our models are not fully autoregressive in nature and do not make any Gaussianity assumptions and instead use a fully neural parameterization of the output probabilities, which contributes to improved performance and modeling flexibility. Approaches like AQF (Si et al., 2022) and NAF (Huang et al., 2018) also provide neural approximators, but their fully autoregressive architecture makes for slow sampling. Closely related feedforward models include GMMNets (Li et al., 2015; Dziugaite et al., 2015) and CramerGANs (Bellemare et al., 2017), but they do not offer likelihood evaluation and posterior inference. Our framework is closely related to Maximum Mean Discrepancy (MMD) (Gretton et al., 2008) and its generalizations (Li et al. (2015), Dziugaite et al. (2015)). However, unlike models optimizing MMD (Li et al., 2015; Dziugaite et al., 2015), ours provide latent variable inference and exact density evaluation using the change of variables formula.

**Conclusion**   We proposed training normalizing flows using the energy objective, a proper scoring rule that does not require computing determinants during training. We then proposed semi-autoregressive energy flows, which feature fast sampling and high sample quality. Using an invertible flow architecture also allows us to retain exact posterior inference in energy flows. We see our work as questioning the use of likelihood for training normalizing flows, and as a first step towards determinant-free flows based on novel learning objectives and architectures.

## 7 REPRODUCIBILITY

**Datasets**  All datasets that we use are open-source. We have described in detail the datasets used in our experiments. The UCI dataset is used without preprocessing, and the digits and MNIST datasets are cast to floats and scaled to [0, 1] range from their original range.

**Models**  For each of our baselines as well as the main code, we have given implementational details in the Appendix A including layers, learning rate, and epochs. For our SAEF, we additionally provide a pseudocode in Appendix F.

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

Table 8: Abbreviations for models (upper) and losses (lower) used throughout the text.

| Abbreviation | Full Name |
|---|---|
| *Models* | |
| AQF | Autoregressive Quantile Flow |
| DIF | Dense Invertible Flow |
| MAF | Masked Autoregressive Flow |
| REF | Rectangular Flow |
| SAEF | Semi-Autoregressive Energy Flow |
| iResNet | Invertible Residual Network |
| Pixel | PixelCNN |
| *Losses* | |
| E | Energy Loss |
| LL | Log Likelihood |
| QL | Quantile Loss |

## A  ADDITIONAL EXPERIMENTAL DETAILS

### A.1  ARCHITECTURE DETAILS

**Abbreviations and Losses**  We have included a table of abbreviations in Table 8, where the upper half consists of the abbreviation for the model names, and the bottom half consists of the abbreviations for losses.

Throughout the text we use loss abbreviations appended to model abbreviations to indicate which objective a particular model was trained with (e.g., PixelMAF-LL corresponds to a PixelMAF model trained with log likelihood). For the remaining baseline models which do not follow this convention (FFJORD, GLOW, iResNet) we note that they are trained with log-likelihood, and note that Flow-GAN uses an adversarial loss combined with a log likelihood term.

**Slicing**  For our slicing experiments with the various multidimensional and single-dimension losses, the invertible feedforward model consists of 4 layers of size 43 (to match Miniboone dimension), and each objective is trained for 200 epochs with a learning rate of 1e-3. Each sample is projected onto 200 random normal vectors, and the resulting two-sample loss is summed across the 200 projections.

**UCI Experiments**  For the PixelMAF and PixelAQF models, the LSTM architectectures are composed of two LSTM layers with hidden size equal to the dimensionality of the data. All models were trained with a batch size of 200 and learning rate of 1e-3 using the Adam optimizer for 200 epochs for the smaller datasets (Miniboone, Hepmass) and 20 epochs for the larger ones (Gas, Power, BSDS 300).

**Image Generation Experiments**  For the PixelCNN architecture, we used a receptive field 7 for digits, and a receptive field of 15 for MNIST. We chose to switch the autoregressive architectures here from LSTM based models which are more sequential to the convolutional PixelCNN architecture to retain local features. The PixelMAF-LL and PixelMAF-QL autoregressively transforms pixels from the image into a distribution of the next pixel, except that PixelMAF-LL uses log likelihood loss while PixelMAF-QL uses quantile loss.The PixelAQF-QL model takes in samples from a uniform distribution, and predicts the corresponding quantile loss for each pixel.

Our DIF model are parametrized with a layer size of 64 on the digits dataset, and 784 on MNIST, with invertible activation functions (Leaky ReLU activations up until the last layer, and Sigmoid for the final activation). The images are flattened out prior to being passed through the feedforward model. On MNIST, each model was trained for 300 epochs with a learning rate of 1e-3, while on digits, each model was trained for 2000 epochs.

The additional baselines are constructed as such: Each model is adapted from the code from their original papers. FFJORD models use two blocks of stacked CNF layers composed of ODE Nets. We

tested with both a hidden size of 64 and a hidden size of 16, and trained for a total of 100 epochs with the optimizer at a learning rate of 1e-3. The Invertible ResNet has three scale blocks, each having 32 Invertible ResNet blocks, consisting of 32 filters with three convolution types with an ELU activation function (Behrmann et al., 2019). It used an AdaGrad optimize with a batch size of 128 trained for 70 epochs at a learning rate of 0.003. Glow was trained with 3 levels and a depth of 1 on 8 GPUs for 250 epochs.

## A.2   D-Loss

The D-Loss is derived from an ensemble of SVM discriminators (with RBF kernels having a bandwidth parameter $\gamma$ of 0.0001, 0.001, 0.01, 0.1, 1, 10, 100, 1000, and 10000) used to differentiate the real data versus the sampled data from the model. More details are included in the appendix. The D-Loss for a single discriminator model $D : \mathbb{R}^n \to \{0, 1\}$ is measured as the accuracy with which D can discern the difference between generated samples and true samples, which are labeled 0 and 1 respectively. Given validation set with $m$ data points and labels $Y \in \{0, 1\}$, we have that

$$\text{D-Loss}_\gamma = \frac{1}{m} \sum_i^m \mathbf{D}_\gamma(x^{(i)}) y^{(i)} + (1 - \mathbf{D}_\gamma(x^{(i)}))(1 - y^{(i)})$$

The final D-Loss is measured by the validation accuracy achieved by the best SVM, which is gotten through a 80:20 train-validation split on the 300 real and 300 fake images: $\text{D-Loss} = \max_\gamma \text{D-Loss}_\gamma$

## A.3   DEFINING CRPS AND U-CRPS

For simulated samples $X_1, ..., X_m$ and true data $y$, we define CRPS as

$$\text{CRPS}((X_1, ..., X_m), y) = \frac{1}{m} \sum_{i=1}^m ||X_i - y||^2 - \frac{1}{2m^2} \sum_{i=1}^m \sum_{j=1}^m ||X_i - X_j||^2$$

and U-CRPS as

$$\text{U-CRPS}((X_1, ..., X_m), y) = \frac{1}{m} \sum_{i=1}^m |X_i - y| - \frac{1}{2m^2} \sum_{i=1}^m \sum_{j=1}^m |X_i - X_j|$$

## B   THEORETICAL ANALYSIS AND PROOFS

**Multi-Variate Objectives**   In this section, we make the assumption that the kernels $K$ used to define our objectives are measurable and bounded by $\kappa$. Under these conditions, when using a kernelized objective with kernel $K$, each distribution $F$ can be represented by a mean embedding $\mu_F$ in the reproducing kernel Hilbert space (RKHS) induced by $K$ (Gretton et al., 2008). We also assume that there exists a unique mapping between $\mu_F$ and $F$ for every $F$ in the class of model distributions $\mathcal{F}$. Note that this can be satisfied for any Borel probability measure if the kernel $K$ is chosen to be universal or characteristic (Gretton et al., 2008). Alternatively, we may satisfy this claim by choosing our $\mathcal{F}$ to be a restricted set of distributions for which the above claim is true.

**Theorem.** *The energy objectives (2) and (3) are consistent estimators for the data distribution and feature unbiased gradients.*

*Proof.* First, we seek to establish the consistency of the minimize of our objectives as an estimator of the data distribution. Our argument uses the fact that the (kernelized) energy score is closely connected to the maximum mean discrepancy (MMD; (Gretton et al., 2008)). Observe that

$$
\begin{aligned}
\mathbf{E}_{y' \sim \mathcal{D}} \text{CRPS}_K(F, \mathbf{y}') &= \frac{1}{2} \mathbb{E}_F K(\mathbf{Y}, \mathbf{Y}') - \mathbb{E}_{F, \mathcal{D}} K(\mathbf{Y}, \mathbf{y}') \\
&= \left( \frac{1}{2} \mathbb{E}_F K(\mathbf{Y}, \mathbf{Y}') - \mathbb{E}_{F, \mathcal{D}} K(\mathbf{Y}, \mathbf{y}') - \frac{1}{2} \mathbb{E}_{y, y' \sim \mathcal{D}} K(\mathbf{y}, \mathbf{y}') \right) + \frac{1}{2} \mathbb{E}_{y, y' \sim \mathcal{D}} K(\mathbf{y}, \mathbf{y}') \\
&= -\frac{1}{2} \text{MMD}_K^2(F, \mathcal{D}) + \text{const};
\end{aligned}
$$

hence, by maximizing the CRPS over a set of possible models $F$, we are minimizing a monotonic transformation of the MMD. Since (2) is a special case of (3) with the distance kernel (Sejdinovic et al., 2013), the above claim also holds for the non-kernelized energy objective (2).

We would like to establish that minimizing objectives (2) and (3) over a data distribution $\mathcal{D}_n$ of $n$ samples from the true data distribution $\mathcal{P}$ yields a model $F_n$ that is similar to what we would obtain if we searched for the best $F$ using the full data distribution $\mathcal{P}$; in other words:

$$\mathbb{E}[L(F_n, \mathcal{P})] \leq \inf_{F \in \mathcal{F}} L(F, \mathcal{P}) + o(n),$$

where $L(F, \mathcal{P})$ is a metric or pseudo-metric[1] that we will instantiate shortly, $\mathcal{F}$ is the hypothesis class for the model $F$, $F_n$ is the empirical risk minimization solution (from our method) over a dataset $\mathcal{D}_n$, and the additive $o(n)$ term decays to zero as we increase $n$. Note that if the model is well-specified (i.e., $\mathcal{P} \in \mathcal{F}$), we have $\mathbb{E}[L(F_n, \mathcal{P})] = o(n)$, and we have a consistent estimator.

To establish this fact, we will derive a version of the above identity for a modified version of the MMD and under the assumption of this section; we will also argue that the kernelized energy estimate satisfies that identity.

First, let $L(F, \mathcal{P}) = \text{MMD}(F, \mathcal{P})$. By the properties of MMD and kernels, we know that $\text{MMD}(F, \mathcal{P}) = ||\mu_F - \mu_{\mathcal{P}}||_{\mathcal{H}}$, and MMD is a pseudo-metric. Note that we have by the triangle inequality

$$L(F_n, \mathcal{P}) \leq L(F_n, \mathcal{D}_n) + L(\mathcal{D}_n, \mathcal{P}),$$

where we overload notation and use $\mathcal{D}_n$ to also denote the empirical distribution. Note that because our objective is a monotonic transformation ($\frac{1}{2}\text{MMD}^2 + \text{const}$) of the MMD, the $F_n$ minimizes the MMD within $\mathcal{F}$. Thus we can write for any $F \in \mathcal{F}$

$$L(F_n, \mathcal{P}) \leq L(F, \mathcal{D}_n) + L(\mathcal{D}_n, \mathcal{P})$$
$$\leq L(F, \mathcal{P}) + 2L(\mathcal{D}_n, \mathcal{P})$$

where we have used once more the triangle inequality in the last line. Taking expectations on both sides and using the fact that $F \in \mathcal{F}$ was arbitrary, we find that

$$\mathbb{E}L(F_n, \mathcal{P}) \leq \inf_{F \in \mathcal{F}} L(F, \mathcal{P}) + 2\mathbb{E}L(\mathcal{D}_n, \mathcal{P})$$
$$\leq \inf_{F \in \mathcal{F}} L(F, \mathcal{P}) + 2\sqrt{\mathbb{E}L(\mathcal{D}_n, \mathcal{P})^2}.$$

To establish our claim, we need to bound the last term. Let $x_i$ denote the i.i.d. samples from $\mathcal{D}_n$, let $\phi$ denote the embedding induced by the kernel $K$ in its RKHS $\mathcal{H}$, and note that we have

$$\mathbb{E}L(\mathcal{D}_n, \mathcal{P})^2 = \mathbb{E}||\frac{1}{n}\sum_{i=1}^{n} \phi(x_i) - \mathbb{E}\phi(x)||_{\mathcal{H}}$$
$$= \text{Var}(\frac{1}{n}\sum_{i=1}^{n} \phi(x_i))$$
$$= \frac{1}{n}\text{Var}(\phi(x_1))$$
$$\leq \frac{2}{n}\mathbb{E}||\phi(x_1)||_{\mathcal{H}}$$
$$\leq \frac{2\kappa}{n}$$

Thus, our main claim follows with

$$\mathbb{E}L(F_n, \mathcal{P}) \leq \inf_{F \in \mathcal{F}} L(F, \mathcal{P}) + 2\sqrt{\frac{2\kappa}{n}}.$$

---

[1]A metric $d(x, y)$ satisfies four properties: symmetry, triangle inequality, $d(x, x) = 0$ and $d(x, y) = 0 \iff x = y$. A pseudo-metric satisfies only the first three properties. The MMD objective is a metric if its kernel is characteristic or universal (or more generally if there is a one-to-one mapping between $\mu_F$ and $F$); otherwise, it is a pseudo-metric.

Thus the estimated model $F_n$ satisfies the above inequality and if the data distribution $\mathcal{P} \in \mathcal{F}$, our consistency claim holds.

We can establish that the gradients are unbiased by leveraging properties of proper scoring rules. Recall from the background section that a loss $L : \Delta(\mathbb{R}^d) \times \mathbb{R}^d \to \mathbb{R}$ is strictly proper (Gneiting and Raftery, 2007a) if $G = \arg\min_F \mathbb{E}_{\mathbf{y} \sim G} L(F, \mathbf{y})$. In the context of the CRPS objective $L$, we have by definition of a proper loss

$$L(F, G) = \mathbf{E}_{y' \sim G} \mathrm{CRPS}_K(F, \mathbf{y}') = \mathbf{E}_{y' \sim G} \mathrm{CRPS}_K(F, G_{\mathbf{y}'}),$$

where $G_{\mathbf{y}'}$ is the empirical distribution derived from $\mathbf{y}'$.

Let $\mathcal{P}$ denote the true data distribution, $\mathcal{D}_n$ a dataset of size $n$ drawn from $\mathcal{P}$, and $G_n$ the resulting empirical distribution. Then we have:

$$\begin{aligned}
\nabla_\theta \mathbb{E}_{\mathcal{D}_n \sim \mathcal{P}} L(F_\theta, G_n)) &= \nabla_\theta \mathbb{E}_{\mathcal{D}_n \sim \mathcal{P}} \mathbb{E}_{\mathbf{y}' \sim \mathcal{D}_n} L(F_\theta, \mathbf{y}') \\
&= \nabla_\theta \mathbb{E}_{\mathbf{y}' \sim \mathcal{P}} L(F_\theta, \mathbf{y}') \\
&= \nabla_\theta L(F_\theta, \mathcal{P}),
\end{aligned}$$

which is equivalent to the statement that we wanted to prove.

$\square$

**Alternative Approaches to Showing Consistency**    The fact that consistency holds also follows from properties of the MMD for general classes $\mathcal{F}$; for example as shown in Dzuigaite et al. (Dziugaite et al., 2015) (Theorem 1),

$$\mathbb{E}[\mathrm{MMD}^2(F_n, \mathcal{P})] \leq \inf_{F \in \mathcal{F}} \mathrm{MMD}^2(F, \mathcal{P}) + o(n),$$

if $\mathcal{F}$ satisfies a fat-shattering condition. The desired consistency claim with $L(F, G)$ being our kernelized energy objective $\mathrm{CRPS}_K$ then follows directly from our earlier derivation by applying an affine transformation on each side of the above equation.

**Alternative Approaches to Showing that Gradients are Unbiased**    Note that a special case of the unbiased gradient property for the non-kernalized objective (1) has been established using techniques discussed in Bellemarre et al. (Bellemare et al., 2017) (Proposition 3). This result also follows from our aforementioned connection to the MMD and Lemma 6 in Gretton et al. (2012) (Gretton et al., 2008).

**Sliced Objectives**

**Theorem.** *The sliced versions of the energy objectives (2) and (3) are consistent estimators for the data distribution and feature unbiased gradients.*

*Proof.* We establish the first part of the claim by observing that the sliced version of the energy objective

$$\mathrm{CRPS}(F, \mathbf{y}') = \mathbb{E}_{w \sim p(w)} \left[ \frac{1}{2} \mathbb{E}_F K(w^\top \mathbf{Y}, w^\top \mathbf{Y}') - \mathbb{E}_F K(w^\top \mathbf{Y}, w^\top \mathbf{y}') \right], \tag{6}$$

where $K$ is a kernel in 1D, is an affine transformation of squared MMD. Then the first part of the claim follows by the argument in Theorem 1. To see this, first, define the function $K_w : \mathbb{R}^d \times \mathbb{R}^d \to \mathbb{R}$ as

$$K_w(x, y) = K(w^\top x, w^\top y),$$

where $K$ is the kernel used as part of the sliced energy objective. It is easy to see that $K_w(x, y)$ is a kernel. Consider any dataset $S = \{x_i\}_{i=1}^k$; then the matrix $M$ defined as $M_{ij} = K_w(x_i, x_j)$ will be semi-definite because the corresponding matrix $M'$ defined as $M'_{ij} = K(w^\top x_i, w^\top x_j)$ is also positive definite, because it is the kernel matrix for the set $S = \{w^\top x_i\}_{i=1}^k$. Hence, by Mercer's theorem $K_w$ is a kernel.

Next, define the function $\bar{K} : \mathbb{R}^d \times \mathbb{R}^d \to \mathbb{R}$ as

$$\bar{K}(x, y) = \mathbb{E}_{w \sim p(w)} K_w(x, y).$$

This is also a kernel, because it is a sum of kernels. Next, note that

$$\text{CRPS}(F, \mathbf{y}') = \mathbb{E}_{w \sim p(w))} \left[ \frac{1}{2} \mathbb{E}_F K(w^\top \mathbf{Y}, w^\top \mathbf{Y}') - \mathbb{E}_F K(w^\top \mathbf{Y}, w^\top \mathbf{y}') \right]$$

$$= \frac{1}{2} \mathbb{E}_F \bar{K}(\mathbf{Y}, \mathbf{Y}') - \mathbb{E}_{F,\mathcal{D}} \bar{K}(\mathbf{Y}, \mathbf{y}'),$$

which is an instance of the kernelized energy objective that uses a modified kernel. Note that this is both a proper score and a rescaled version of the squared MMD with a modified kernel. The two claims of this theorem follow directly from Theorem 1.

$\square$

## C  EXPANDED BACKGROUND ON PROPER SCORING RULES

### C.1  PREDICTIVE UNCERTAINTY IN MACHINE LEARNING

Probabilistic machine learning models predict a probability distribution over the target variable—e.g. class membership probabilities or the parameters of an exponential family distribution. We seek to produce models with accurate probabilistic outputs that are useful for generation.

**Notation.**   Supervised models predict a target $y \in \mathcal{Y}$ from an input $x \in \mathcal{X}$. , where $x, y$ are realizations of random variables $X, Y \sim \mathbb{P}$ and $\mathbb{P}$ is the data distribution. We are given a model $H : \mathcal{X} \to \Delta_{\mathcal{Y}}$, which outputs a probability distribution $F(y) : \mathcal{Y} \to [0, 1]$ within the set $\Delta_{\mathcal{Y}}$ of distributions over $\mathcal{Y}$; the probability density function of $F$ is $f$. We are also given a training set $\mathcal{D} = \{(x_i, y_i) \in \mathcal{X} \times \mathcal{Y}\}_{i=1}^n$ consisting of i.i.d. realizations of random variables $X, Y \sim \mathbb{P}$, where $\mathbb{P}$ is the data distribution.

### C.2  PROPER SCORING RULES

Comparing point estimates from supervised learning models is straightforward: we can rely on metrics such as accuracy or mean squared error. Probabilities, on the other hand, are more complex and require specialized metrics.

In statistics, the standard tool for evaluating the quality of predictive forecasts is a proper scoring rule (Gneiting et al., 2007). This paper advocates for evaluating the quality of uncertainties using proper scoring rules (Gneiting and Raftery, 2007b).

Formally, let $L : \Delta_{\mathcal{Y}} \times \mathcal{Y} \to \mathbb{R}$ denote a loss between a probabilistic forecast $F \in \Delta_{\mathcal{Y}}$ and a realized outcome $y \in \mathcal{Y}$. Given a distribution $G \in \Delta_{\mathcal{Y}}$ over $y$, we use $L(F, G)$ to denote the expected loss $L(F, G) = \mathbb{E}_{y \sim G} L(F, y)$.

We say that $L$ is a *proper loss* if it is minimized by $G$ when $G$ is the true distribution for $y$: $L(F, G) \geq L(G, G)$ for all $F$. One example is the log-likelihood $L(F, y) = -\log f(y)$, where $f$ is the probability density or probability mass function of $F$. Another example is the check score for $\tau \in [0, 1]$:

$$\rho_\tau(F, y) = \begin{cases} \tau(y - F^{-1}(\tau)) & \text{if } y \geq f \\ (1 - \tau)(F^{-1}(\tau) - y) & \text{otherwise.} \end{cases} \tag{7}$$

See Table 9 for additional examples.

What are the qualities of a good probabilistic prediction, as measured by a proper scoring rule? It can be shown that every proper loss decomposes into a sum of the following terms (Gneiting et al., 2007):

$$\text{proper loss} = \text{calibration} \underbrace{- \text{sharpness} + \text{irreducible term}}_{\text{refinement term}}.$$

Thus, there are precisely two qualities that define an ideal forecast: calibration and sharpness.

| Proper Loss | Loss $L(F, G)$ | Calibration $L_c(F, Q)$ | Refinement $L_r(Q)$ |
|---|---|---|---|
| Logarithmic | $\mathbb{E}_{y \sim G} \log f(y)$ | $\mathrm{KL}(q \| f)$ | $H(q)$ |
| CRPS | $\mathbb{E}_{y \sim G} \left(F(y) - G(y)\right)^2$ | $\int_{-\infty}^{\infty} (F(y) - Q(y))^2 \mathrm{dy}$ | $\int_{-\infty}^{\infty} Q(y)(1 - Q(y)) dy$ |
| Quantile | $\mathbb{E}_{y \sim G}^{\tau \in U[0,1]} \rho_\tau(F, y)$ | $\int_0^1 \int_{Q^{-1}(\tau)}^{F^{-1}(\tau)} (Q(y) - \tau) dy d\tau$ | $\mathbb{E}_{y \sim Q}^{\tau \in U[0,1]} \rho_\tau(Q, y)$ |

Table 9: Examples of three proper losses: the log-loss, the continuous ranked probability score (CRPS), and the quantile loss. A proper loss $L(F, G)$ between distributions $F, G$—assumed here to be cumulative distribution functions (CDFs)—decomposes into a calibration loss term $L_c(F, Q)$ (also known as reliability) plus a refinement term $L_r(Q)$ (which itself decomposes into a sharpness and an uncertainty term). Here, $Q(y)$ denotes the CDF of $\mathbb{P}(Y = y \mid F_X = F)$, and $q(y), f(y)$ are the probability density functions of $Q$ and $F$.

# D DETAILS ON ADDITIONAL TWO-SAMPLE TRAINING OBJECTIVES

## D.1 GAUSSIAN TWO-SAMPLE BASELINE OBJECTIVE OVER HIGH-DIMENSIONAL VECTORS

We use the following classical objectives as baselines for our work and to illustrate examples of alternative methods that can be derived from our two-sample-based approach; both tests make a Gaussian modeling assumption.

**Hotelling's Two-Sample Test** Being closely related to Student's t-test, it uses the following statistic:

$$H_2(\mathcal{D}_F, \mathcal{D}_G) = (\mathbf{m}_F - \mathbf{m}_G)^\top S^{-1}(\mathbf{m}_F - \mathbf{m}_G), \tag{8}$$

where $\mathbf{m}_F = \frac{1}{m} \sum_{\mathbf{y}^{(i)} \in \mathcal{D}_F} \mathbf{y}^{(i)}$, $\mathbf{m}_G = \frac{1}{m} \sum_{\mathbf{y}^{(i)} \in \mathcal{D}_G} \mathbf{y}^{(i)}$ are the sample means, and the matrices $S_F = \frac{1}{m-1} \sum_{\mathbf{y}^{(i)} \in \mathcal{D}_F} (\mathbf{y}^{(i)} - \mathbf{m}_F)(\mathbf{y}^{(i)} - \mathbf{m})^T$, $S_G = \frac{1}{m-1} \sum_{\mathbf{y}^{(i)} \in \mathcal{D}_G} (\mathbf{y}^{(i)} - \mathbf{m}_G)(\mathbf{y}^{(i)} - \mathbf{m})^T$ are sample covariances, while $S = (S_F + S_G)/2$ is their average. This objective encourages the two samples to have similar means.

**Fréchet Distance** This is another Gaussian-based distance that we use as an objective:

$$R(\mathcal{D}_F, \mathcal{D}_G) = ||\mathbf{m}_F - \mathbf{m}_G||_2^2 + \mathrm{tr}(S_F + S_G - 2(S_F S_G)^{1/2}), \tag{9}$$

where we are using the same notation as above. This objective is encouraging the model to produce data with similar means and variances. It is derived from the Fréchet distance between two Gaussians.

## D.2 SLICED TWO-SAMPLE BASELINES

Slicing also allows us to use univariate two-sample tests as objectives. We give examples below.

**Kolmogorov-Smirnov** One of the most popular ways of comparing the similarity between two distributions is via the quantity

$$\mathrm{KS}(F, G) = \sup_y |F(y) - G(y)|, \tag{10}$$

the maximum distance between two CDFs $F$ and $G$. Empirical CDFs can be used for samples. While this test corresponds to an IPM, it does not have a widely accepted extension to higher dimensions.

**Hotelling's Univariate Objective** The sliced version of Hotelling's objective corresponds to using Hotelling's $t^2$ univariate test (which is just the squared version of Student's $t$-test) as an objective.

$$H_u(\mathcal{D}_F, \mathcal{D}_G) = \frac{(m_F - m_G)^2}{s^2}, \tag{11}$$

where $m_F, m_D$, and $s^2$ are respectively the sample mean of $\mathcal{D}_F$, the sample mean of $\mathcal{D}_G$, and the combined sample variance, defined as in the multivariate version. Note that this formula is much less computationally expensive than the multi-dimensional one, which requires performing a matrix inversion (in worst-case $O(d^3)$ time), while the sliced version takes only $O(d)$ time.

**Fréchet Univariate Objective**    Similarly, the sliced version of the Fréchet objective is written as:

$$R_u(\mathcal{D}_F, \mathcal{D}_G) = (m_F - m_G)^2 + (s_F^2 - s_G^2)^2, \tag{12}$$

which encourages the sample means $m_F, m_G$ and the sample variances $s_F, s_G$ to be the same. Again, this $O(d)$ formula is less computationally expensive than in higher-dimensions, where it requires performing multiple matrix multiplications and a matrix square root ($O(d^3)$).

## E    Feed-Forward Architectures for Energy Flows.

Since our objective does not require computing Jacobians, we are able to use flexible classes of invertible models that are difficult to train using log-likelihood. Previous work on invertible mapping leveraged integration-based transformers (Wehenkel and Louppe, 2021), spline approximations (Müller et al., 2019; Durkan et al., 2019; Dolatabadi et al., 2020), piece-wise separable models, and others. Our main requirement on the model architecture is efficient sampling. We describe several feed-forward flow architectures that are compatible with our objective below.

**Dense Invertible Layers**    The simplest architecture we consider consists of a sequence of small $\mathbb{R}^d \to \mathbb{R}^d$ dense layers with invertible non-linearities (such as tanh or leaky ReLUs). We enforce the invertibility of the dense layers by adding a scaled identity component $\sigma I_d$ for small $\sigma > 0$; other options for inducing invertibility include positivity constraints on the weights (Huang et al., 2018). Although the tanh non-linearities are invertible, numerical values close to $\{-1, 1\}$ tend to introduce numerical instability during inversion; we address this issue by via activity regularization (Chollet et al., 2015). With these two architectural choices, we were able to compute both $z \to y$ and $y \to z$ mappings analytically and in a numerically stable way for modestly sized $d$'s; we provide experiments below.

**Invertible Residual Networks**    Recently, residual networks with spectral normalization have been proposed as a flexible invertible architecture (Behrmann et al., 2019). Although one of the two directions of the flow is not computable analytically, it may be approximated using a fixed-point iteration algorithm. Invertible residual networks are typically trained using maximum likelihood; computing the determinant of the Jacobian of each layer requires a sophisticated approximation based on Taylor series expansion. Interestingly, when training these flows using maximum likelihood, the "fast" direction needs to be $y \to z$ in order to enable fast training, but generation becomes non-analytic; when training using an energy objective, the $z \to y$ direction is fast both for training and generation.

**Rectangular Flow Architectures**    Recently, several authors explored rectangular flows, in which the dimensionality of $y$ and $z$ is not equal (Nielsen et al., 2020; Cunningham and Fiterau, 2021; Caterini et al., 2021). Training with maximum likelihood involves sophisticated extensions to the change of variables formula; however, these models can be trained without modification using an energy loss as long as one can sample from them efficiently. At the same time, we may retain their pseudo-invertibility to perform posterior inference.

## F    Pseudocode for Semi-Autoregressive Flows

Training:

Sampling:

## G    Motivation for Energy Flows

Flows have a somewhat broader set of benefits and motivations:

1. Exact training using maximum log-likelihood (as opposed to e.g., ELBO).

2. Accurate (exact) latent inference, i.e. obtaining "good" z from x. Note: This is a major challenge in generative models: it motivates complex inference frameworks in GANs (e.g., BiGAN, ALI

---

**Algorithm 1** Semi-Autoregressive Energy Flow Training

1: **for** $iteration = 1, 2, \ldots$ **do**
2:     Generate $\mathbf{z}$ of random normals
3:     **for** $b = 1, 2, \ldots, Blocks$ **do**
4:         $\mathbf{h}_b = c_b(\mathbf{x}_{<b})$
5:         $\mathbf{y}_b = \tau(\mathbf{z}_b; \mathbf{h}_b)$
6:         $\mathbf{y}'_b = \tau(\mathbf{z}'_b; \mathbf{h}_b)$
7:         Compute Energy Loss$(\mathbf{y}_b, \mathbf{y}'_b, \mathbf{x}_b)$
8:     **end for**
9:     Backpropagate sum of energy loss on $\tau$
10: **end for**

---

**Algorithm 2** Semi-Autoregressive Energy Flow Training

    **for** $b = 1, 2, \ldots, Blocks$ **do**
2:     $\mathbf{h}_b = c_b(\mathbf{y}_{<b})$
        $\mathbf{y}_b = \tau(\mathbf{z}_b; \mathbf{h}_b)$
4: **end for**

---

(Dumoulin et al., 2016; Donahue et al., 2016)), and there is also a line of research on advanced variational posteriors $q(x|z)$ in VAEs (e.g., AAEs, AVB, etc. (Makhzani et al., 2015; Mescheder et al., 2017)).

3. Exact likelihood evaluation, which supports using the likelihood as an evaluation metric (separately from the training objective, which could be different) for principled comparisons between models.

The key point is that switching to energy objectives loses point 1, but preserves points 2 and 3, and also yields the following additional benefits:

4. The ability to use highly expressive invertible parameterizations beyond the ones that are currently compatible with maximum likelihood training. This can help better fit the data distribution and enables fast sampling.

5. Fast training via an objective that does not require computing determinants of Jacobians.

As a result, we think that energy flows are conceptually interesting as a potential improvement over likelihood-based flows. Furthermore, we view flows simply as bijective mappings between a simple and a complex distribution and likelihood is a tool for training flows, not an intrinsic goal or defining feature. By questioning the use of likelihood as an objective, we improve truly important features of flows: expressivity, fast training and inference, exactly inferable latents.

Flows are defined to be bijective, but this often requires compromises on expressivity. Our work shows ways to avoid these compromises while retaining most benefits of flows.

From our experiments in Table 4, both feed-forward and autoregressive flows trained with the energy loss feature poor log-likelihoods. However, their FID and CRPS metrics are excellent: these models still generate very good images. We note that previous work has already shown that models trained with non-likelihood objectives tend to have very weak log-likelihoods; for example in existing work on training normalizing flows with adversarial losses (Grover et al., 2018). Our work adds additional results to this line of work.

It may appear that energy flows do not retain a key benefit of normalizing flows—being able to compare models via their log-likelihood. However, we argue that there are additional nuances. First, even though model comparison using log-likelihood is challenging, energy flows can still be used to study the successes and limitations of log-likelihood as a metric. Additionally, it is unclear when exactly log-likelihood fails for model evaluation—for example, recent diffusion models attain high log-likelihoods despite being trained in practice with an objective that is not a principled lower bound on the likelihood. Thus, there may be other architectures, modified objectives, and use cases for flows where log-likelihood calculations will be useful.

Finally, energy flows still retain a combination of features that is rare among generative models: (1) exact posterior inference; (2) fast feed-forward generation; (3) a stable and effective training

objective; (4) implicitly defined distributions (i.e. p(x|z) is not assumed to be of any parametric form); (5) equal support for continuous and discrete outputs; (6) equal support for continuous and discrete latents. Closest in terms of features to energy flows are Generative Moment Matching Networks (GMMNets (Dziugaite et al., 2015; Li et al., 2015))—our work can be seen as adding a principled way of doing posterior inference in GMMNets (they normally don't support it). Also related are VAEs; however they do not possess properties (4) and (6), and across our experiments generate more blurry samples (we can just swap out the objective and keep the same generator), as evidenced by FID and qualitative inspection. Other families like GANs and autoregressive models have even greater differences in terms of training stability and generation speed respectively.

