# OpenReview forum: "Semi-Autoregressive Energy Flows: Towards Determinant-Free Training of Normalizing Flows"
_ICLR.cc/2023/Conference — Submitted to ICLR 2023_

### Official Review · Reviewer_Ljxv · 2022-10-25

**Confidence:** 4
**Correctness:** 3
**Technical Novelty And Significance:** 3
**Empirical Novelty And Significance:** 2
**Recommendation:** 6

**Clarity, Quality, Novelty And Reproducibility:**

The paper is clearly written and well organized. The method is novel and the argumentation correct. My main concern regarding the quality of the paper is that the experimental evaluation of the method has flaws that I mentioned above.

**Strength And Weaknesses:**

### Strengths

Computing the Jacobian determinant of normalizing flows is a challenge, but is required for most existing training procedures. Often, the architecture is designed such that the computation becomes simple or approximations are used, but this can limit expressivity and accuracy. The authors address this challenge in a creative way by proposing a novel training procedure based on two-sample test objectives.
Furthermore, they demonstrate that such a training procedure unlocks new opportunities in terms of flow design by introducing a novel flow model that can only be trained with their method.
They apply their method to flows of various architectures and, thereby, demonstrate its effectiveness. Important is the runtime comparison, since the superior runtime is one of the main arguments for using their method.

### Weaknesses

My main criticism concerns the experimental evaluation. The authors apply their method to various normalizing flows for which alternative training procedures exist, but the baseline models trained via maximum likelihood have a different architecture. There should also be experiments with the same flow architecture but different training methods being used. Moreover, the use of the CRPS and the NLL as evaluation objectives make it difficult to compare the work with other articles in the field, where the likelihood (UCI) or bits per dim (MNIST) are the main evaluation metrics being used. Computing them might be expensive for some of the models, but this is important information to evaluate the method.

**Summary Of The Paper:**

In this paper, the author's establish a novel method to train normalizing flows without the need to compute the likelihood. Thereby, the computation of the Jacobian determinant of the flow is avoided, which can be expensive or requires a restricted architecture to be cheap to compute. They propose objectives inspired by two-sample tests, such as the energy loss, and show how they can be estimated and used for training. This objective can be used to train a variety of flows, and the authors focus on applying it to flows having Jacobian determinants that are expensive to compute. Moreover, they introduce a novel flow architecture, called semi-autoregressive flow, that can only be trained with their energy loss.
In their experiments, the authors demonstrate that their approach is competitive with other training procedures on UCI and MNIST datasets, while being often faster to train and evaluate.

**Summary Of The Review:**

Due to the importance and the novelty of this method, I tend towards accepting this paper. However, I ask the author to address the concerns I voiced regarding the experimental evaluation

Edit after rebuttal:

I thank the reviewers for their response and additional experiments. I am still inclined to accept the article.

---

> ### Author Response · Authors · 2022-11-18
> **Response to Reviewer Ljxv**
>
> We thank the reviewer for the detailed comments. Below we discuss the reviewer’s main comments:
>
> **Baseline models trained via maximum likelihood have a different architecture**
> We note that the baseline methods we train for MAF share the same architecture as our SAEF models, and we also included other dense models in order to broaden our set of baselines.
>
>
> | Method       | U-CRPS        | CRPS          | D-Loss        | FID           | MMD            | Training (sec) | Sampling (sec) |
> |--------------|---------------|---------------|---------------|---------------|----------------|--------------------------------|--------------------------------|
> | PixelMAF-LL  | .128          | .279          | 1.00          | 100.55        | 35             | 195.38        | 0.296          |
> | PixelMAF-QL  | .099          | .215          | .983          | 85.08         | 35             | 195.38        | 0.287          |
> | PixelAQF-QL  | .119          | .228          | .986          | 61.27         | 29             | 195.38                               |
> | SAEF-2       | .085          | .188          | .675          | 9.86          | 0.167          | 32                             | 31.22                          |
> | SAEF-4       | .085          | .187          | $\textbf{.567}$ | $\textbf{7.05}$ | 0.081          | 12                             | 8.19                           |
> | SAEF-7       | .085          | .187          | .608          | 14.91         | 0.088          | 6                              | 2.17                           |
> | SAEF-14      | .085          | .187          | .650          | 19.57         | 0.068          | 5                              | 0.93                           |
>
>
> Additionally, we included stronger baselines in the form of FFJORD, iResNet, and GLOW (which were adapted from their original implementations). Finally, we also added a Flow-GAN baseline during this rebuttal period (please see updated Table 7, also copied below), which was  trained with the same architecture as our dense invertible flows.
>
> | Method       | U-CRPS        | CRPS          | D-Loss        | FID           | MMD            | Training (sec) | Sampling (sec) |
> |--------------|---------------|---------------|---------------|---------------|----------------|--------------------------------|--------------------------------|
> | FFJORD (16)  | .101          | .208          | .650          | 24.78         | 0.103          | 540                            | 48.88                          |
> | FFJORD (64)  | .102          | .209          | .633          | 9.69          | 0.087          | 3100                           | 155.69                         |
> | iResNet      | .100          | .206          | .642          | 41.47         | 0.111          | 840                            | 2.43                           |
> | Flow-GAN     | .085          | .187          | 0.608         | 43.67         | 0.068          | 15                             | 0.40                           |
> | GLOW         | 0.090         | 0.197         | 0.983         | 63.06         | 0.600          | 1400                           | 190.63                         |
> | REF-E        | .085          | .187          | .778          | 41.04         | 0.052          | 3                              | 0.21                           |
> | DIF-E        | $\textbf{.084}$ | $\textbf{.186}$ | .701          | 22.76         | $\textbf{0.051}$ | 6                              | 0.40                           |
> | DIF-E-Proj   | .085          | .186          | .819          | 22.55         | 0.056          | 3                              | 0.40                           |
>
>
> **Reporting NLL and bits per dim metrics for SAEF & Baselines**
>
> We are showing NLL numbers for various types of generative models (notably, energy flows) in Table 4, also reproduced below. Our model was trained with CRPS and has significantly worse NLL, however we find that this is completely uncorrelated with image quality, as seen by our models’ competitive FID scores.
>
> This observation demonstrates that NLL is generally uncorrelated from sample quality, and does not need to be used as a training objective. Our findings lend further support for using likelihood-free objective for generative models.
>
> | Models      | NLL   | CRPS | FID    |
> |-------------|-------|------|--------|
> | PixelMAF-LL | -3142 | .279 | 100.15 |
> | PixelMAF-QL | -2617 | .215 | 85.08  |
> | DIF-E       | 2388  | .186 | 22.76  |

---

### Official Review · Reviewer_B3hX · 2022-10-26

**Confidence:** 3
**Correctness:** 3
**Technical Novelty And Significance:** 2
**Empirical Novelty And Significance:** 2
**Recommendation:** 3

**Clarity, Quality, Novelty And Reproducibility:**

Novelty:
  - Using IPMS to train normalizing flow has been explored before [1].
  - The use of slicing seems interesting, but it isn't clear if this was used in the experiments. Furthermore, this seems similar in spirit to using random features for kernel estimation [3].

Clarity:
  - This paper discusses multiple objectives for training. Which objective(s) were used for training the models in the experiments? Were the baselines trained with the same objective (CPRS) or maximum likelihood?
  - I feel like NLL values should be reported in Table 5, since all of these are invertible and can compute log-likelihood values.
  - The NLL values should be negated in Table 4.

Comments:
  - Can the inverse of SAEF be computed? Perhaps not analytically, but an iterative procedure [4] should work.
  - SAEF's log probability can be computed by computing the determinants of each block (which I assume is how the NLL estimates are computed in Table 4). Have the authors tried training with maximum likelihood and comparing it to the sample-based objectives?

[3] "Random Features for Large-Scale Kernel Machines." Ali Rahimi and Ben Recht. (2007)

[4] "Mintnet: Building invertible neural networks with masked convolutions" Song et al. (2019)

**Strength And Weaknesses:**


Weakness:
  - The use of IPM are standard in the GAN literarture. From what I understand, the main thing separating this work from those training invertible generative models with a discriminator (e.g. [1]), is that there are no parameters in this objective. In fact, the CPRS objective (Eq 3) has the same gradient as MMD, so the only reason this isn't adversarial is because unlike the MMD-GAN objective, this work does not make use of a neural network-defined kernel function.
  - Related to the above, the only experiments are in tabular and MNIST, where I imagine simple kernels and distance metrics suffice. But for training on more complex distributions like colored images, a neural network discriminator is crucial.
  - Furthermore, this work does not convincingly show that training without a discriminator is sufficient, since Flow-GAN [1] is not compared against.
  - In terms of evaluation metrics, it would be better to use an evaluation metric that is different from the training objective for SAEFs, which heavily biases towards SAEF. Have the authors considered metrics such as MMD, NLL, or a kernel-based density estimator [2]?

[1] "Flow-GAN: Combining Maximum Likelihood and Adversarial Learning in Generative Models." Grover et al. (2018)

[2] "On the Quantitative Analysis of Decoder-Based Generative Models." Wu et al. (2016)

**Summary Of The Paper:**

This paper proposes training normalizing flows using sample-based objectives.

**Summary Of The Review:**

Since the paper focuses on sample quality and sample-based metrics, I feel the experiments on only tabular and MNIST are not sufficient. Furthermore, I'm not convinced this is better than the existing variety of GAN objectives such as MMD-GAN. The main difference seems to be the lack of a neural network for defining a kernel function; however, human-designed metrics tend to scale poorly to more complex data sets.

---

> ### Author Response · Authors · 2022-11-18
> **Response to Reviewer B3hX**
>
> We thank the reviewer for the detailed feedback. Below we address the main points raised by the reviewer.
>
> **Experiments on colored images**
> While we believe that the results we report for UCI datasets and MNIST in terms of improved training time and performance are strong, we agree with the reviewer that CIFAR-10 experiments would make our experiments more compelling. We intend to use this discussion period to run experiments on CIFAR-10. We look forward to discussing those results with you as we gather them.
>
> **Comparison to IPMs in Normalizing flows (e.g., Flow-GAN)**
> We have added two additional baselines to our experimental results on MNIST (Table 7, copied below for reference), one of which is Flow-GAN. SAEF outperforms both of these models in terms of FID and runtime. We also further note that our use of the energy objective for normalizing flows is motivated by the desire to capture the benefits of both the normalizing flow and generative adversarial approaches. That is, GANs do not require any expensive log determinant calculations but have empirically proven to be unstable to train leading to collapsed solutions. In contrast, normalizing flows are more stable but require an expensive log determinant calculation. Our proposed method leverages the stability of normalizing flows without incurring the expensive calculations that are entailed in computing the log likelihood objective function.
>
> | Method       | U-CRPS        | CRPS          | D-Loss        | FID           | MMD            | Training (sec) | Sampling (sec) |
> |--------------|---------------|---------------|---------------|---------------|----------------|--------------------------------|--------------------------------|
> | FFJORD (16)  | .101          | .208          | .650          | 24.78         | 0.103          | 540                            | 48.88                          |
> | FFJORD (64)  | .102          | .209          | .633          | 9.69          | 0.087          | 3100                           | 155.69                         |
> | iResNet      | .100          | .206          | .642          | 41.47         | 0.111          | 840                            | 2.43                           |
> | $\textbf{Flow-GAN}$     | .085          | .187          | 0.608         | 43.67         | 0.068          | 15                             | 0.40                           |
> | GLOW         | 0.090         | 0.197         | 0.983         | 63.06         | 0.600          | 1400                           | 190.63                         |
> | REF-E        | .085          | .187          | .778          | 41.04         | 0.052          | 3                              | 0.21                           |
> | $\textbf{DIF-E}$        | $\textbf{.084}$ | $\textbf{.186}$ | .701          | 22.76         | $\textbf{0.051}$ | 6                              | 0.40                           |
> | DIF-E-Proj   | .085          | .186          | .819          | 22.55         | 0.056          | 3                              | 0.40                           |
> | SAEF-2       | .085          | .188          | .675          | 9.86          | 0.167          | 32                             | 31.22                          |
> | $\textbf{SAEF-4}$       | .085          | .187          | $\textbf{.567}$ | $\textbf{7.05}$ | 0.081          | 12                             | 8.19                           |
> | SAEF-7       | .085          | .187          | .608          | 14.91         | 0.088          | 6                              | 2.17                           |
> | SAEF-14      | .085          | .187          | .650          | 19.57         | 0.068          | 5                              | 0.93                           |
>
>
> **Reporting additional metrics**
> We thank the reviewer for this suggestion. We have added the MMD metric in Table 7 (as shown above) for each of the models. We find that models trained with the energy objective yield low MMD, and in fact the DIF-E model attains the lowest MMD.
>
> **Clarifying use of slicing in experiments**
> Slicing is a technique used to improve computational efficiency. Our introduction of slicing in the context of energy losses was to demonstrate that these losses can benefit from the scalability improvements of slicing while maintaining consistency and unbiased gradients and without sacrificing performance. Table 3 (as shown below) reports our slicing experiment results, where we find that performance for sliced energy objectives are comparable to their non-sliced counterparts.
>
> | n       | 400   | 200   | 100   | 50    |
> |---------|-------|-------|-------|-------|
> | U-CRPS  | 0.088 | 0.088 | 0.088 | 0.091 |
> | CRPS    | 0.191 | 0.191 | 0.192 | 0.195 |
>
>
> | Block-n | 100   | 20    | 10    | 5     |
> |---------|-------|-------|-------|-------|
> | U-CRPS  | 0.084 | 0.085 | 0.084 | 0.084 |
> | CRPS    | 0.086 | 0.087 | 0.086 | 0.087 |

---

> > ### Author Response · Authors · 2022-11-18
> > **Continuation**
> >
> > **Clarifying which objective is used for training each model**
> > We have added the abbreviations for the losses in Table 8 in Appendix A.1 with additional content in that section detailing losses for each model:
> >
> > **Training SAEF with Log Likelihood**
> > We agree with the reviewer that this is an interesting experiment that will further illuminate the benefits of energy flows. We are currently working on conducting this experiment and look forward to sharing and discussing the results with you during the discussion period.
> >
> > **Negating values for NLL in Table 4**
> > The numbers in Table 4 are correct. Our model was trained with CRPS and has worse NLL, however we find that this is completely uncorrelated with image quality, as seen by our models’ competitive FID scores.
> >
> > **Invertibility of Energy Flows**
> > We include negative log-likelihood for the non-autoregressive version of energy flows in Table 4 (as shown below), so indeed energy flow models are invertible. We also leverage invertibility in Figure 1, where we show smooth interpolation between intermediate digit representations for the DIF-E model.
> >
> > | Models      | NLL   | CRPS | FID    |
> > |-------------|-------|------|--------|
> > | PixelMAF-LL | -3142 | .279 | 100.15 |
> > | PixelMAF-QL | -2617 | .215 | 85.08  |
> > | DIF-E       | 2388  | .186 | 22.76  |

---

### Official Review · Reviewer_gK13 · 2022-11-04

**Confidence:** 3
**Clarity, Quality, Novelty And Reproducibility:** See my comments above.
**Correctness:** 3
**Technical Novelty And Significance:** 3
**Empirical Novelty And Significance:** 2
**Recommendation:** 5

**Strength And Weaknesses:**

# Strengths
- The idea presented in this paper is exciting and somehow novel. It efficiently trains invertible architectures while keeping the potential advantage of their invertibility at test time. The generalisation of the ideas from Si et al. (2022) to non-autoregressive architectures is well-exposed and has a practical interest.
- The empirical results demonstrate that using the energy metric to train the flow is a good objective function if one is interested in good sample qualities (although it is not always clear how we define good samples) or by the marginal densities (as well represented by the CRPS score).

# Weaknesses
Although I think the ideas presented in the paper make sense and could have a real interest for some practical use cases, I find the paper's organisation and way of presenting ideas confusing. For instance, the tables' order does not follow the structure of the paper and the captions are uninformative, which makes it hard for the reader to follow the results at first glance. In addition, the discussions in section 3 go in many different directions and are missing a link between them. It is only at a second read that we can more or less grasp what each paragraph brings to the paper.

I am also a bit confused by the argument from the authors that autoregressive architectures lead to "good sample quality". They can be good indeed. Still, I do not see why they are better than other generative models in terms of sample quality.

The authors say that MAF leads to gaussian distribution, which is only valid for a 1-step flow but is immediately relaxed if there are dependencies between variables and one uses more than one step of the flow. Indeed, MAF allows non-linear interaction between the variables. Thus the corresponding joint and marginal distribution can be non-Gaussian.

I also think the author should mention that CRPS only reflects the quality of the marginal distribution and cannot say whether the model captures dependencies well.

At some point, the authors say that using the sliced objective (which I find a neat idea) reduces the computational complexity. Of course, this is true. However, it is unclear to which extent this is necessary, I would have liked to see numbers for that.

I recommend that the authors create a better structure and story for this paper. I do think the ideas are excellent and (probably) well-motivated theoretically. However, the article in its current form is not pleasant to read, which I find sad for the quality of the ideas presented.



**Summary Of The Paper:**

This paper presents a novel strategy to train invertible neural networks that model a distribution, a.k.a. normalizing flows. In particular, the authors propose to use proper metrics as the objective function to train flows. This approach relaxes the requirement of computing the determinant of the Jacobian of the flow and unlocks architecture for which computing this determinant would be too computationally demanding.

Authors strongly motivate their approach with theoretical arguments for the soundness of proper metrics. They also validate their approach experimentally, showing improvements in the sample quality compared to likelihood-based training.

**Summary Of The Review:**

Although the idea and experiments presented in this paper are interesting. I find the presentation too poor to argue for acceptance. Hence I argue for a weak reject.

---

> ### Author Response · Authors · 2022-11-18
> **Response to Reviewer gK13**
>
> We thank the reviewer for recognizing the novelty and potential impact of our proposal. Below we address the main discussion points that the reviewer raised.
>
> The primary impetus for our work was to find an alternative to the maximum likelihood paradigm in training normalizing flows which requires either expensive log determinant calculations or compromises on model architectures and expressivity. We proposed the novel use of energy objectives as a principled alternative to maximum likelihood training of normalizing flows, which circumvents the need for expensive log-determinant calculations. Our empirical results demonstrate that this alternative objective does not sacrifice performance and in some cases attains the best quality of generated samples compared to similar and even more expressive baseline models trained with log likelihood. More generally, it questions the use of log likelihood as a training objective for generative models, similar to existing work on adversarial training objectives (e.g., FlowGAN); unlike that work it provides a stable alternative to adversarial training.
>
> **MAF joint distribution clarification**
> For MAFs, we only stated that the conditionals are gaussian; the joint can indeed be non-gaussian, but their flexibility is still limited.
>
> **“CRPS only reflects the quality of the marginal distribution and cannot say whether the model captures dependencies well.’’**
> Although this statement is true for the univariate CRPS (U-CRPS), we also calculated and reported multidimensional CRPS, which does indeed capture the dependencies between dimensions.
>
> **Use of sliced objectives**
> Our introduction of slicing in the context of energy losses was to demonstrate that these losses can benefit from the scalability improvements of slicing while maintaining consistency and unbiased gradients and without sacrificing performance. Table 3, shown below, reports our slicing experiment results, where we find that performance for sliced energy objectives are comparable to their non-sliced counterparts.
>
> | n       | 400   | 200   | 100   | 50    |
> |---------|-------|-------|-------|-------|
> | U-CRPS  | 0.088 | 0.088 | 0.088 | 0.091 |
> | CRPS    | 0.191 | 0.191 | 0.192 | 0.195 |
>
>
> | Block-n | 100   | 20    | 10    | 5     |
> |---------|-------|-------|-------|-------|
> | U-CRPS  | 0.084 | 0.085 | 0.084 | 0.084 |
> | CRPS    | 0.086 | 0.087 | 0.086 | 0.087 |
>
> The computational savings are detailed in Table 2, provided below, where we see that slicing provides significant improvements when the number of projections $n$ is much smaller than the dimensionality of the data $d$.
>
> | Metrics    | KS         | 1D Hotelling | Hotelling | 1D-FD | FD    | 1D-Energy | Energy |
> |------------|------------|--------------|-----------|-------|-------|-----------|--------|
> | CRPS       | 1.53       | 0.57         | 0.717     | 0.558 | 0.559 | 0.545     | 0.548  |
> | Complexity | $n\log{b}$ | $n$          | $d^3$     | $n$   | $d^3$ | $bn$      | $bd$   |
>
>
> **Table ordering**
> Thank you for identifying this issue. We have fixed the table ordering in our updated manuscript.

---

### Official Review · Reviewer_WiBH · 2022-11-04

**Confidence:** 4
**Correctness:** 3
**Technical Novelty And Significance:** 3
**Empirical Novelty And Significance:** 3
**Recommendation:** 3

**Clarity, Quality, Novelty And Reproducibility:**


The paper is reasonably easy to follow, at least on a high level. The paper is very heavy on the use of acronyms, which makes it a little difficult to follow for one who is not intimately familiar with all of these acronyms and their variations. A section on each acronym used in the tables and a brief description of them would be helpful. Also, in the case of CRPS vs U-CRPS, a mathematical definition would be useful as opposed to just a textual description.

Minor nitpicks:
- Table 5 should be lower down, the current draft has it above Table 4.
- Appendix typo: "the above **clam** also holds"

**Strength And Weaknesses:**

Strengths
- The energy loss is straightforward, faster to compute, and produces good samples.
- Slicing provides an effective way of scaling the training.
- Semi-autoregressive flows elegantly trade-off computation and expressivity.
- Exact posterior calculation is possible, allowing for things like interpolation that is not easy to do in autoregressive models.

Weaknesses
- The experiments are only done on UCI and handwritten digit datasets. At the very least, CIFAR-10 would have been good.
- Comparisons to larger-scale flow models like GLOW would be more convincing.
- There are key differences between the behavior of the energy loss and NLL (they are not strictly correlated). More exploration of where they differ in terms of allocation of probability mass would be nice.

**Summary Of The Paper:**

This paper uses the energy loss for training normalizing flows. This produces a consistent estimator that doesn't require computing determinants of the Jacobian of the transformation, making it more flexible in terms of the choice of architecture. The paper also introduces a few tricks to further improve scaling and/or performance: slicing the data using 1D projections, and a semi-autoregressive flow model that interpolates between a full autoregressive model and a completely parallel model (i.e., a typical normalizing flow). Experiments on UCI and handwritten digit datasets show favorable results compared to baseline flows, VAEs, and autoregressive models, while allowing for fast sampling and exact posterior inference.

**Summary Of The Review:**

The use of the energy loss is compelling, and the background behind it is nicely written. This approach frees normalizing flows from their previous architectural constraints, and this paper explores this idea through the introduction of a semi-autoregressive flow model that interpolates between faster sampling and better sample quality.

I think that the ideas certainly have merit, however it feels like there is a bit of a gap between the promise of the paper and the actual demonstration of the approach. Specifically, the experiments are very small-scale for 2022, only operating on UCI datasets and MNIST. I'm surprised that CIFAR-10 wasn't included, as it is commonly used as a baseline for generative models. For example, one method referenced and compared with (Papamakarios et al., 2017) uses CIFAR-10, and it is now 5 years old. Normalizing flows (e.g., GLOW),  PixelCNN, and VAEs have all been scaled to more challenging datasets many years ago.

I certainly appreciate a careful and detailed study of the estimator and its effects on small-scale data, however the conspicuous lack of larger-scale experiments tell me that either the authors have not yet run these experiments, or there is a severe limitation that is keeping the model from working on those datasets.

I personally appreciate new and elegant ideas, and so all I'm asking for is an attempt on something a more challenging baseline. If the results are negative, then a discussion about reasons for this, and avenues for improvement would be necessary. The intention is to ensure that this paper is carried forward and does not languish because other researchers find it cannot scale well.

If there is a strong reason why the existing experiments are sufficient, then I would be happy to discuss this with the authors.

Minor comments:
I think you should add additional references for normalizing flows.
Laurent Dinh, David Krueger, Yoshua Bengio, "NICE: Non-linear Independent Components Estimation", ICLR 2015
Laurent Dinh, Jascha Sohl-Dickstein, Samy Bengio, "Density estimation using Real NVP", ICLR 2017

---

> ### Author Response · Authors · 2022-11-18
> **Response to Reviewer WiBH**
>
> We thank the reviewer for the detailed feedback and suggestions. Below we summarize the main points and minor comments with our responses inline.
>
> ### Main feedback
> **Adding colored images experiments**
> While we believe that the results we report for UCI datasets and MNIST in terms of improved training time and performance are strong, we agree with the reviewer that CIFAR-10 experiments would make our experiments more compelling. We intend to use this discussion period to run experiments on CIFAR-10. We look forward to discussing those results with you as we gather them.
>
>
> **Adding comparison to other large scale models, e.g., Glow**
> We have added two additional baselines to our experimental results on MNIST (Table 7): Glow and Flow-GAN. SAEF outperforms both of these models in terms of FID and runtime.
>
> | Method       | U-CRPS        | CRPS          | D-Loss        | FID           | MMD            | Training (sec) | Sampling (sec) |
> |--------------|---------------|---------------|---------------|---------------|----------------|--------------------------------|--------------------------------|
> | FFJORD (16)  | .101          | .208          | .650          | 24.78         | 0.103          | 540                            | 48.88                          |
> | FFJORD (64)  | .102          | .209          | .633          | 9.69          | 0.087          | 3100                           | 155.69                         |
> | iResNet      | .100          | .206          | .642          | 41.47         | 0.111          | 840                            | 2.43                           |
> | Flow-GAN     | .085          | .187          | 0.608         | 43.67         | 0.068          | 15                             | 0.40                           |
> | GLOW         | 0.090         | 0.197         | 0.983         | 63.06         | 0.600          | 1400                           | 190.63                         |
> | REF-E        | .085          | .187          | .778          | 41.04         | 0.052          | 3                              | 0.21                           |
> | DIF-E        | $\textbf{.084}$ | $\textbf{.186}$ | .701          | 22.76         | $\textbf{0.051}$ | 6                              | 0.40                           |
> | DIF-E-Proj   | .085          | .186          | .819          | 22.55         | 0.056          | 3                              | 0.40                           |
> | SAEF-2       | .085          | .188          | .675          | 9.86          | 0.167          | 32                             | 31.22                          |
> | SAEF-4       | .085          | .187          | $\textbf{.567}$ |$\textbf{7.05}$ | 0.081          | 12                             | 8.19                           |
> | SAEF-7       | .085          | .187          | .608          | 14.91         | 0.088          | 6                              | 2.17                           |
> | SAEF-14      | .085          | .187          | .650          | 19.57         | 0.068          | 5                              | 0.93                           |
>
>
> ### Additional comments
> **Adding an acronym definition section**
> Thank you for this useful suggestion. We have added this to Appendix A.1. (Table 8).
>
> **Mathematical definition of CRPS vs. U-CRPS**
> Thank you for this suggestion. We have added this definition to Appendix A.3.
>
> **Move Table 5 lower down**
> We have moved Table 5.
>
> **Appendix typo: "the above clam also holds"**
> Thank you for finding this typo. We have fixed it.
>
> **Additional references**
> Thank you for these suggested references. We have added them to our paper

---

> > ### Comment · Reviewer_WiBH · 2022-12-09
> > **Still waiting on CIFAR-10 results**
> >
> > Thank you to the authors for your detailed response. I am willing to raise my score by one point to acknowledge the additional baselines and clarifications.
> >
> > However, while you have addressed most of my concerns, you haven't yet addressed the most important one: a larger-scale experiment (ideally on CIFAR-10). You had mentioned that you would spend the discussion period working on this experiment, I'm wondering if you have any updates on this?
> >
> > Thank you,
> >
> > Reviewer WiBH

---

### Author Response · Authors · 2022-11-18
**Changes To Manuscript**

We thank all the reviewers for their detailed feedback. We have posted responses to each reviewer's individual comments. Here, we detail all the changes that we have made to our manuscript in light of this initial discussion period:
* Added Flow-GAN and Glow baselines to Table 7 (MNIST generation).
* Added MMD evaluation to Table 7 (MNIST generation).
* Added a formal definition of CRPS and U-CRPS to Appendix A.3.
* Added acronyms in Table 8 and a clarification text on which objectives were used to train which models in Appendix A.1.
* Added citations.
* Re-ordered tables.
* Fixed minor typos.

---

### Decision · Program_Chairs · 2023-01-20

**Decision:**

Reject

**Justification For Why Not Higher Score:**

The paper does not include sufficient experimental validation to validate the premise of the proposed approach.

**Justification For Why Not Lower Score:**

N/A

**Metareview: Summary, Strengths And Weaknesses:**

This paper proposes to use energy loss for training normalizing flows, avoiding the computation of determinants of the Jacobian of the transformation.

Main strengths:
- Well-motivated approach for using energy loss for training normalizing flows.

Main weakness:
- The experiments do not sufficiently validate the promise of the paper. The evaluated datasets are of a very small scale. Even results from CIFAR-10 are not included.

The reviewers appreciate the authors' responses but still find the missing larger-scale evaluation concerning.

The AC agrees with the reviewers that the paper is not yet ready for publication. The AC thus recommends to reject.